# Drought Impacts on the Crop Sector and Adaptation Options in Burkina Faso: A Gender-Focused Computable General Equilibrium Analysis

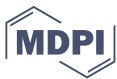

**Boureima Sawadogo**

EDEHN-Equipe d'Economie Le Havre Normandie, Le Havre Normandy University, 76600 Le Havre, France; tboureima94@yahoo.fr

**Abstract:** The crop sector in Burkina Faso has been facing recurrent droughts since 1970. This study analyzes the impacts of droughts and adaptation options such as the use of irrigation capacity development methods, integrated soil management and drought-tolerant crop varieties on the crop sector. Indeed, we focus on the consequences of agricultural droughts on economic growth and employment and on the gender dimension of household poverty. Using a gendered dynamic computable general equilibrium model linked to a microsimulation model, we conduct simulations of various drought scenarios (mild, moderate and severe). We show that the mild, moderate and intense droughts experienced by Burkina Faso over the past ten years have negatively affected the country's economic performance and considerably degraded the welfare of its households. The gross domestic product has fallen by 3.0% in the short term and 3.3% in the long term due to intense droughts. Moreover, the number of poor people is growing faster in male-headed households than in female-headed households. Given the large female population in both groups of households, women bear the brunt of droughts. These results also demonstrate how these negative impacts could worsen in the future with the recurrence of intense droughts due to the threat of global climate change. We find that Burkina Faso has room to reduce the negative impacts by adopting drought-tolerant crop varieties, integrated soil management approaches or expanding their irrigation capacity.

**Keywords:** CGE model; agricultural drought; gender; poverty; Burkina Faso

## 1. Introduction

It is widely recognized that climate change is one of the most important global issues of the twenty-first century. According to the statements made in the IPCC reports [1], the consensus among the scientific community has grown stronger regarding the increase in temperature across the planet. In addition, climate change is also manifested by changes in precipitation and humidity and increases in tropical cyclones, with effects on droughts, floods and ecosystems [2,3]. Regarding droughts, changes in temperature and precipitation are expected to alter their frequency and intensity. The existing evidence shows that droughts have a negative impact on rural households in developing countries [4–7]. Moreover, other studies underline that female-headed households are most vulnerable to these extreme weather events in the developing world [8–10]. Recurrent droughts have reduced GDP growth in many African countries [11–13] and threatened development prospects [14–16].

Droughts are the most regular meteorological phenomena in Burkina Faso [17,18]. Since the 1970s, Burkina Faso has experienced chronic drought, with the most critical phases being the years 1973–1974 and 1983–1984 [17]. Moreover, since 1991, the country has experienced about seven major droughts, with the most recent ones being in 2011–2012, 2014 and 2020. Indeed, it appears that past droughts have resulted in production losses [19]. In addition, the consequences of production losses include increased food prices, food shortages and reduced food consumption, with consequences for malnutrition [20–22]. For

the 2011–2012 drought, for example, farmers lost an estimated 25–75% of their farm income and their livestock income [23]. During 2011–2012, due to the drought, the number of undernourished people in the country increased from 3.8 million (2008–2010) to 4.4 million (2011–2013), or about one-quarter of the population [23]. In the same period, the drought left 450,000 children under the age of five in a situation of severe malnutrition and 100,000 in a situation of acute malnutrition [23]. According to the International Water Association, between 1969 and 2014, droughts cumulatively affected 12.4 million people [24]. Future projections show probable increases in temperature of 1.4–1.6 °C by the year 2050 [25] and a rainfall reduction of 7.3% by 2050 [26]. In addition to the frequency of the droughts, the length of the dry season and also the length of the periods of low rainfall tend to exacerbate the effects. However, the uneven temporal and partial distribution of rainfall makes it difficult to assess the intensity of droughts across the country in order to provide the country with policies and measures to adapt to droughts.

However, droughts affect all countries, but poor countries are reportedly the most severely affected [27]. Among the populations of these countries, the most vulnerable are the most affected [28]. Indeed, vulnerable populations such as women are the most affected by droughts, as they are more often employed in the agricultural sector [28–30]. Although the impacts of drought on agricultural production are well documented, little is known about the exact consequences on gender disparities. Indeed, in Burkina Faso, economic disparities could be reinforced and prolonged as the country has experienced since 2016 an armed conflict, since 2020 the coronavirus pandemic (COVID-19) and also in 2022 the effects of the Russo–Ukrainian conflict. However, in this study, even though the country faces multidimensional crises that create very serious socio-economic situations, we focus on the effects of the climate crisis.

This paper aims to assess the impacts of agricultural drought and adaptation options on growth, focusing on rates of employment and poverty based on gender. There is ample historical evidence that droughts have caused agricultural production losses and the migration of pastoralists from their original agroecological zones in Burkina Faso [19,31,32]. Burkina Faso's agriculture is mainly rainfed, and this low-income country is particularly vulnerable to drought. In addition to the direct effects on agriculture, livestock and farming and herding households, droughts also affect the non-agricultural sectors as well as non-farming households. The droughts of the 1970s, 1980s and 2011–2012 have shown that droughts also affect the non-agricultural sectors and urban households through increased prices and reduced incomes. The data and evidence show that droughts in recent years have led to poor and vulnerable rural populations migrating to valleys and small basins and to cities [33]. In addition, declining crop yields have impacted food availability and raised concerns about food and nutrition security. Although the effects of drought are visible through soil degradation, water scarcity and the destruction of pastures, the potential magnitude of the impacts of drought in terms of growth and food and non-food poverty are not as well known. Additionally, the existing literature highlights the facets of the different levels of vulnerability and exposure to climate change impacts based on gender, indicating that women are not specifically more exposed, but the interactions between gender, power dynamics, socioeconomic structures and societal expectations mean that the climate impacts are experienced very differently by women [34]. In Burkina Faso, despite progress being made, inequalities based on gender and economic status persist [35].

However, gender disparities in terms of income are the most significant. For example, the gross national income per capita is unequally distributed, at 1336 FCFA for women versus 2077 FCFA for men [36]. This inequality is due to the differences in labor market participation, at 75.1% for men and 58.5% for women [36]. According to the report from the 2018 survey on social institutions and gender equality in Burkina Faso, the distribution of the sectoral participation rates for men and women in the labor market changes across the agriculture sector (73.3% of women against 67.7% of men), industry sector (3.2% of women against 5.7% of men) and service sector (23.5% of women against 23.5% of men) [37]. However, the high labor force participation rate hides high unemployment (affecting 3.0%

of men versus 4.9% of women) and high underemployment rates (16.3% of men versus 30.3% of women) [38]. The agriculture sector is the engine of the economy, contributing 35% of the gross domestic product (GDP) and employing 86% (more than 50% are women) of the economically active population. It provides 61.5% of the monetary income for rural households, which represent 70% of the total population, of which 47.5% are poor. The agricultural production is mainly rainfed, yet it faces cyclical drought episodes that are intensifying with climate change.

In this paper, we investigate specifically the impacts of drought and adaptation options on production, GDP, food security and household-level poverty. However, economic assessments of the impacts of droughts are scarce. The existing studies are partial equilibrium studies [39,40] and only focus on either direct or indirect impacts. In addition, these studies have focused on damage assessments, repair costs, or impacts on market prices. Indeed, a relevant literature review on the methods used for economic evaluations of drought effects [41] showed that general equilibrium methods are the appropriate tools. These tools have the capacity to take into account the sectoral independencies of the whole economy. In addition, they allow for the interrelationships between economic agents (households, government, firms and the rest of the world). However, economywide studies are scarce. For example, [16] evaluated the impacts of drought on the economy of Malawi and found that the gross domestic product (GDP) has declined and poverty among urban and non-farm households has increased. Furthermore, [15] studied the impacts of agricultural drought on the economy of Mali and found that drought episodes degrade economic performance and household welfare. Another study [42] evaluated the impacts of the 2002–2003 drought on the Australian economy and found that the drought reduced the GDP and contribute to a decline in unemployment and to a worsening of the trade balance. Another study [43] assessed the effects of drought on the Mexican economy and concluded that the impacts vary substantially by sector, and also found that adaptation policies can only cause modest changes to the economic losses to be suffered. In the existing literature, the analyses are either ex post (historical approach) studies to assess the impacts of historical events ([42,44]) or ex ante (hypothetical approach) studies to assess hypothetical or future phenomena [15,16]. Indeed, none of the studies has analyzed the impacts of droughts on gender disparities, while droughts could increase existing economic inequalities between men and women.

With regard to the direct and indirect effects of agricultural drought, we use a dynamic recursive computable general equilibrium (CGE) model incorporating specifications for the significant effects of droughts on the agricultural sectors in Burkina Faso. The model is an appropriate and relevant tool because it allows us to compare the results of agroclimatic shocks at the economy level more than other tools, such as the Ricardian model or partial equilibrium model. The CGE model is not limited to the direct effects of shocks on the agricultural sector only, but takes into account the indirect effects through changes in prices, income, supply and demand. In addition, the dynamic specifications allow the generation of time paths of the effects of successive shocks on economic and social variables. In the existing economic literature, a growing number of CGE studies have assessed the impacts of changing weather conditions on the vulnerability of low-income countries with low adaptive capacities. These studies have used two approaches, a deterministic approach to long-term climate conditions (e.g., [45–47]) and a stochastic or probabilistic approach to include variable characteristics of climate conditions (e.g., [15,48–50]). To our knowledge, very little is known about the impacts of drought and adaptation strategies on poverty in developing countries in particular. Among the existing studies, [16,50] coupled the CGE model with a microsimulation model to assess the impacts of drought on poverty. Another knowledge gap in the literature on the economic and social costs of drought is the differentiated impact by gender. Given these gaps in the literature, this study develops a gender-computable general equilibrium model that is linked to a microsimulation module. CGE models are the method of choice for representing the links between different sectors of the economy and agents (households) in a coherent macroeconomic framework. Linking

the CGE model to a microsimulation model will allow us to assess the poverty impacts of drought.

The contributions of this paper are three-fold. First, it presents an economic analysis of the impacts of drought on food security in Burkina Faso, as a case study of a climate-sensitive developing country. Second, it addresses an important issue that is little studied in the literature, as most drought-related papers focus on the economic impacts and neglect the impacts on vulnerable groups. Third, the paper explicitly considers the gender dimension of poverty, a dimension that has been neglected in the literature so far. By showing that the effects of drought are different for male- and female-headed households, it allows for more appropriate policy recommendations to support the most vulnerable. Finally, in this study, we adopt the historical approach and simulate the impacts of various drought events in Burkina Faso over a 22-year period (2018–2040). With the prediction of increased global warming, we focus on three categories of drought, i.e., mild, moderate and intense. Finally, we discuss some drought adaptation strategies in order to identify adaptive strategies that could reduce the adverse effects of droughts in Burkina Faso.

The rest of the paper is organized as follows. Section 2 presents the model, data and simulation scenarios. Section 3 presents the results and discussion, while Section 4 concludes the paper.

## 2. Study Area

Burkina Faso is a Sahelian country in West Africa, situated between the Republic of Mali to the north and west; Niger to the east; Côte d'Ivoire to the southwest; and Ghana, Togo and Benin to the south (Figure 1). The country covers an area about 274 200 km$^2$. It is essentially flat, with an average altitude of about 300 m above sea level [51]. Its climate is characterized by two seasons, a dry season lasting nine months and a rainy season lasting three months. The country is characterized by considerable variations in rainfall, ranging from an annual average of 350 mm in the north to more than 1100 mm in the southwest. The national monthly mean temperature sits between 25.8 °C and 29.6 °C year-round [25]. Burkina Faso can be divided into three agroecological zones (AEZ): 1—Sahelian with insufficient rainfall below 700 mm; 2—the Sudanian zone represents 32.4%, the Sudano–Sahelian zone 38.9% and the Sahelian zone 28.7% of the national territory [52].

Agricultural activities are carried out during the winter season, which runs from June to October each year. Burkina Faso's economy is highly dependent on agricultural production, and about 80% of the country's population is dependent on agriculture, which contributes nearly 28.6% of the national GDP [25]. The agricultural production is mainly concentrated in the southern Sudanian zone, where the climatic conditions are favorable. In addition, the form of agriculture is subsistence and is not very diversified. The export-oriented crops are concentrated on cotton, groundnuts, sesame and rice. However, the most important subsistence crops are sorghum, maize, millet and rice, which account for 62.7% of the total production and constitute the staple foods of Burkina Faso's population. These subsistence crops are mainly rainfed, are grown in areas of less than 5 hectares and provide particularly low yields.

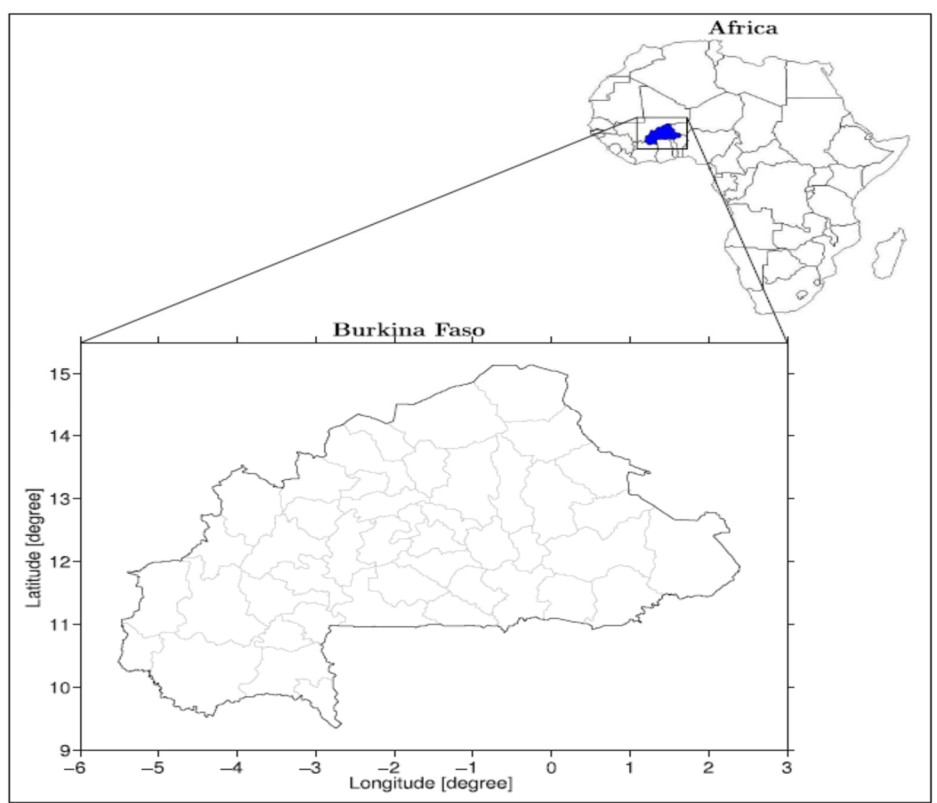

**Figure 1.** Study area and its location in Africa. Source: This figure is taken from [53].

## 3. Model and Data

### 3.1. Macro–Micro Analysis Framework and Data Sources

To understand the economic impacts of droughts on developing economies, several studies have been performed. Most of the existing studies are partial equilibria studies (e.g., [54,55]) that focus on one part of the economy and ignore the indirect effects of drought. To account for the direct and indirect effects of drought events on the economy as a whole, a limited number of studies have employed computable general equilibrium modeling ([15,16,42,50,56–60]).

To assess the implications of the drought on the economy of Burkina Faso as a whole, and in particular in terms of household poverty, we use a CGE model integrated with a poverty analysis model. In the macro–micro coupling, as in [16,50], changes in the consumption expenditure and prices of goods and services calculated in the CGE model are fed into the microsimulation module to determine the household poverty impacts of different characteristics (e.g., gender dimension of the household head) (Figure 2). This coupling allows for a detailed analysis of the economy-wide effects of drought at the sectoral and subnational levels, with an analysis of the poverty impacts of drought.

Therefore, the 2018 SAM is used here for the calibration of the CGE model. Thus, the CGE running with the SAM is used to implement drought simulation scenarios and adaptation options. The impacts on the household production, employment, income, prices and consumption expenditure are generated. The changes in the prices of goods and services and consumption expenditure are then used as inputs to the micro model to update the household consumption expenditure from the national household living conditions survey and the national poverty line. Finally, the new poverty indicators are calculated and compared with the baseline situation.

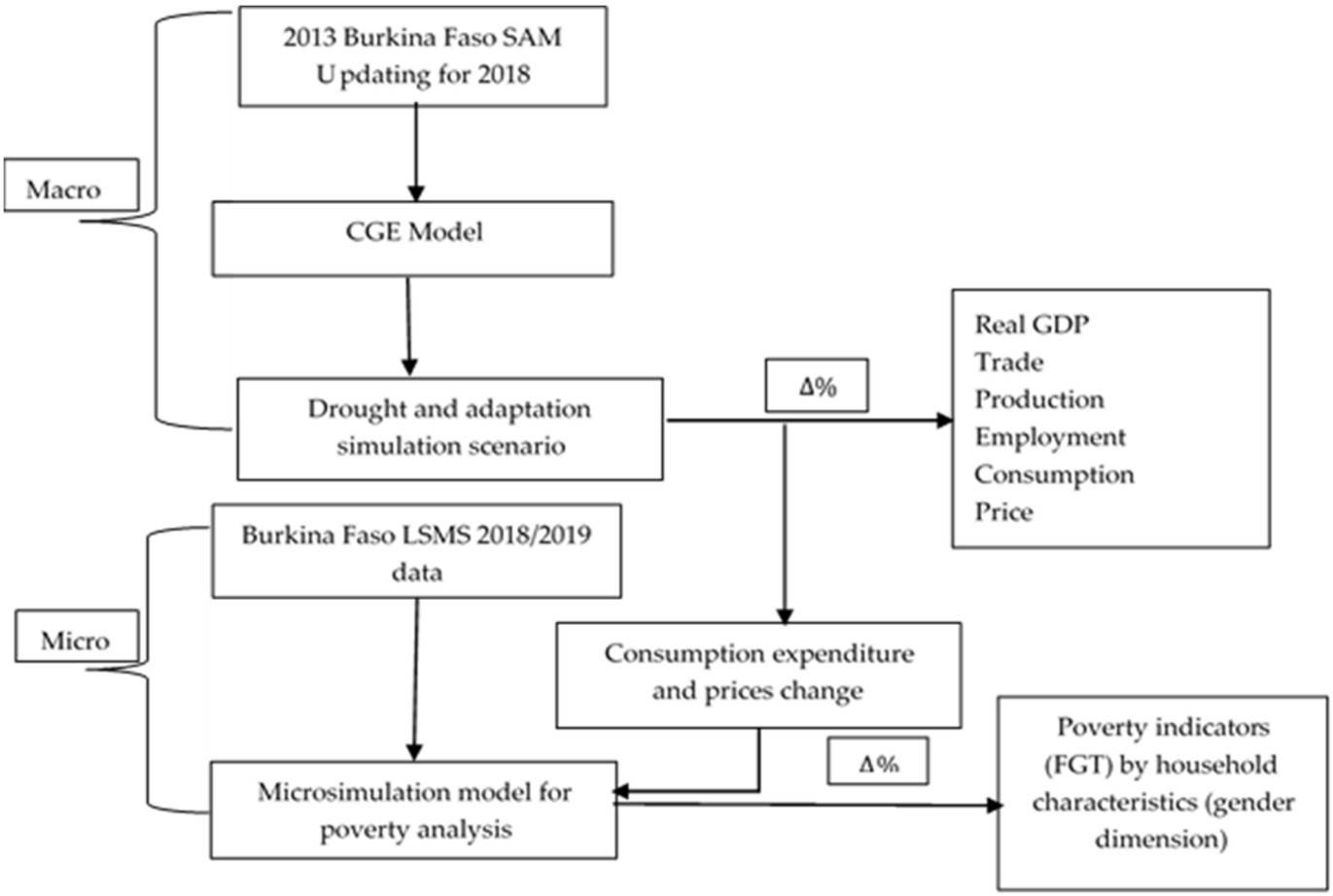

**Figure 2.** Macro–micro coupling framework for poverty analysis. Source: Author's construction.

In order to assess the impacts of the temporal trajectories of drought scenarios on the Burkinabe economy, this study uses a dynamic CGE model based on the PEP 1-t model of [61]. The PEP-1-t CGE model is a popular model, as it is already applied for the analysis of several policies and exogenous shocks for different countries. More recently, for example, [62] used the CGE PEP 1-t model to assess the impact of the COVID-19 pandemic state of emergency and the government's fiscal plan on South Africa's economy and environment. Additionally, [63] with the CGE PEP 1-t model assessed the impact of agricultural sector reforms in Senegal. Although the model is fully described in [61], we present the main assumptions. The CGE model is calibrated with the 2013 Burkina Faso Agricultural Social Accounting Matrix (SAM) [64] updated for the year 2018 with 2020 World Bank development indicator data [65]. Related to the SAM, our model has 27 production sectors, including 9 agricultural activities and 29 products including 10 agricultural products.

The core of the constructed model is based on the neoclassical general equilibrium paradigm. Producers maximize their profit under a given technology and independent prices. Therefore, each industry's representative producers face a nested structure of production. The production is presented in a four-level process (see Figure 3). The Figure 3 shows the steps in the production modelling process. It shows how factors are combined to achieve the level of output in each sector. At the first level, the production function takes a Leontief form for the value added and total intermediate inputs in the fixed share. In other words, the aggregate inputs are considered to be strictly complementary, following a Leontief production function. At the second level, the value added takes a CES (constant elasticity of substitution) from comprising composite labor and composite capital. It is after this second step that our model is different to the PEP model.

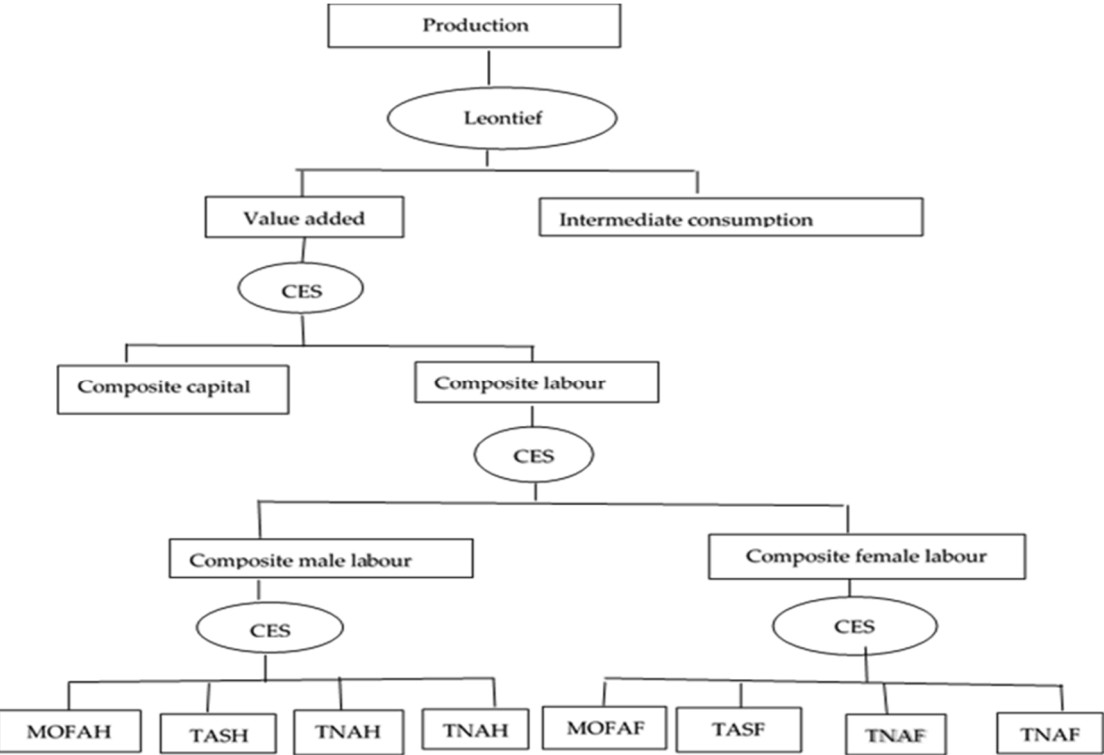

**Figure 3.** Economic production process. Source: Author's construction. Note: MOFAH: family farm labor force male; MOFAF: female family farm labor force; TASH: salaried agricultural work for men; TASF: salaried agricultural work for women; TNAHQ: non-agricultural skilled work for men; TNAFQ: skilled female non-farm work.

Following the social accounting matrix (SAM), labor is disaggregated between agricultural salary labor, family labor and non-farm labor. To take into account the gender dimension of Burkina Faso's economy, the different labor input categories are subdivided by gender. Thus, salaried agricultural labor and family labor are disaggregated by sex. On the other hand, non-agricultural labor is distinguished by sex and the level of qualification. To integrate this disaggregation into our modelling, we assume that at the third level composite labor is a CES function between total agricultural labor and total non-agricultural labor, although it is not relevant to substitute farm labor for non-farm labor. At the fourth level, the total agricultural labor is a CES function between agricultural labor categories (female and male family labor and female and male agricultural salary labor). At the same level, the total non-agricultural labor is a CES function combination between non-agricultural labor categories (female and male skilled labor and female and male unskilled labor).

At the composite capital demand side, the agricultural sectors are all highly agriculturally capital-intensive, and the non-agricultural sectors are also highly non-agriculturally capital-intensive. Thus, the composite capital demand is a Leontief production function between the total agricultural capital and total non-agricultural capital, where the composite capital is directly determined by the stock of the capital (proportional relationship) and there are no substitution possibilities between the total agricultural capital and total non-agricultural capital. Then, the total agricultural capital is a CES function between the agricultural capital categories (land, agricultural equipment) and the total non-agricultural capital is a CES function between the non-agricultural capital categories.

To incorporate the effects of drought into the CGE model, we modify some parameters of the standard PEP-1-t model. The first modification concerns the introduction of the potential effects of drought into agricultural activities. Thus, for a given period t, we assume that the total factor productivity parameter of the value-added function depends

on an exogenous annual growth rate, reflecting a neutral technical change in line with Hicks. This annual growth rate is also a function of a random effect when drought occurs.

The model has four different types of agents. First, households are disaggregated into rural poor households, rural rich households, urban rich households and urban poor households. In this study, we grouped the households into two categories, the rural and the urban households. Each category of household receives capital and labor income and transfers from other institutions. Households pay direct taxes to the government and spend their disposable income by consuming and saving. Household consumption, as a function of prices and income, which is allocated across different commodities, is based on an LES (linear expenditure system) demand function, which is derived from the maximization of a Stone–Geary utility function. A firm's income consists, on the one hand, of its share of capital income, and on the other hand on the transfers received from the other agents. They pay dividends to the different institutions, pay direct taxes, and save money. The revenue for the government comes from direct taxes from households and firms and indirect taxes on activities and commodities. It makes transfers to other agents, buys commodities and saves money. Finally, the rest of the global income comes from its sales on the Burkinabe market, income from labor and capital and transfers from other institutions. It buys commodities and makes transfers to domestic institutions. The difference between the rest of the global spending and income is the current account balance.

On the supply side, the domestic production is either sold on the domestic market or outside of it. It is assumed that there exists a CES function to link the domestically produced goods that are consumed at home with exported goods. A CET (constant elasticity of transformation) function determines the scope of the choices between domestic supply and exports. On the demand side, consumers can either buy commodities that are produced domestically or imported. Their choice will be influenced by the relative prices of domestically produced and imported commodities, as well as the elasticity of the substitution between the imported and domestic commodities.

In terms of the closure rules, we assume that the nominal exchange rate is the numeraire of the model. Burkina Faso is considered to be a small country, meaning it has no influence on global prices. Thus, the global prices for all commodities are fixed. Moreover, we assume that the current account balance is fixed, and this underscores that Burkina Faso cannot borrow as much as it wants from the rest of the world. The capital is mobile across activities, representing a long-term situation where the economy has time to adjust.

As mentioned before, and although our model is inspired by the PEP1-t model, it departs radically, particularly via the introduction of labor market rigidity. Thus, the assumptions of the full employment of factors, wage flexibility and the equality of labor supply and demand for work by type of work and industry are rejected for the non-agricultural labor market. In our work, we postulate that the non-agricultural labor market is not in equilibrium and that ready-made workers remain unemployed. The concept of unemployment must be used with caution in a country like Burkina Faso, but the fact remains that available workers are unemployed, and this is what we will measure as "unemployment". Several theories have been developed to explain wage rigidity. One study [66] provided an excellent review of some of the theories that have been implemented in CGE models. Among the developed wage rigidity theories, three categories are worth mentioning: the search and matching theory by [67], the efficiency wages theory by [68] and the collective wage bargaining theory by [69]. Like [70], we use the most common framework found in the literature, which is the wage curve introduced by [71,72]. We explicitly model unemployment using a wage curve, i.e., unemployment is negatively related to the real wage rate and we use −0.1 as the wage elasticity borrowed from [73].

The dynamics are introduced through growth in the supply of the production factors. The labor supply is assumed to grow at an exogenous population grow rate. Other variables that increase with the population growth are the current account balance, minimum consumption of commodities in the equation of the LES demands, current government expenditures, public investment by category and public sector, and finally changes in

inventory. The capital stock is equal to the level in the preceding period, with less depreciation plus new investment. The allocation of new private capital between categories and industries follows a modified version of the [74] investment demand specification and varies according to the ratio of the rental rate to the user cost of that capital.

Referring to the Food and Agricultural Organization (FAO), we have four pillars of food security: food availability, food access, utilization and stability [75]. To analyze the impacts of drought on food security, we retain two pillars, namely the availability and access to food. Therefore, the two indicators, i.e., availability and access, are calculated directly with the CGE model. The per capita food availability index is measured by the volume of production per capita in urban and rural areas. The per capita food access index is measured by per capita food consumption in rural and urban areas.

Finally, to assess the drought impacts on poverty, we combine our CGE model with the micro model using a top-down approach. Once the simulations are run with the CGE model, the changes in income and prices are transmitted to the micro module. Poverty is measured using the traditional FGT indicators used by [76]. The change in poverty level is calculated by comparing the poverty rate before and after the occurrence of a drought. The household consumption expenditure data are taken from the 2018/2019 Survey on Household Living Conditions from the Institute of Statistics and Demography of Burkina Faso [77].

### 3.2. Data

The database used for the CGE model is the 2018 SAM, which is an updated SAM from the 2013 SAM of the Ministry of Agriculture and Water Development [64]. The SAM represents the different flows between activities, commodities and institutions, as well as flows between institutions (direct taxes paid, dividends received, etc.) in the national economy for the year 2018. Additional data, such as income and trade elasticities from [78], are used to operationalize the model. Finally, to make the poverty analysis possible, data from the 2018/2019 National Survey on Household Living Conditions in Burkina Faso collected by the National Institute of Statistics and Demography [77] are used.

### 3.3. Simulation Scenario Assumption

In this study, two simulation scenarios are implemented. First, a reference scenario is simulated by updating the constant parameters and exogenous variables from one year to the next. To do so, the population growth rate of Burkina Faso is used to build the reference scenario. In a second step, the drought scenario is simulated and compared with the reference scenario. To define the drought scenario, we refer to what has happened over the last ten years in terms of agricultural yields. The scientific literature distinguishes three types of droughts—a precipitation deficit or meteorological drought, a negative water anomaly or hydrological drought and a soil moisture deficit or agricultural drought. In this work we focus on agricultural drought. The soil moisture is a key indicator of drought conditions, since the soil fertility depends on precipitation and evapotranspiration and also on temperature, as higher temperatures result in higher evapotranspiration. According to the RCP2.6 and RCP6.0 model projections, the average annual soil moisture at one meter in height for Burkina Faso will decrease by 2.5% by 2080 [24]. However, uncertainties exist because not all models predict the same direction of soil moisture change.

Since 1970, Burkina Faso has experienced endemic drought; however, the droughts of 1972–1974 and 1983–1984 were the most severe. According to the RCP6.0 model, the probable range of drought exposure for the national cropland area per year increased from 0.07–3.8% in 2000 to 0.04–16% in 2080 [24]. The very likely range increased from 0.01–12.0% in 2000 to 0.01–29.0% in 2080 [24]. According to [79], the probability of drought occurring within a season ranges from 5% to 40%.

To define our drought scenario, we use the standard rainfall and evapotranspiration index calculated by the Global Drought Monitor for Burkina Faso (https://spei.csic.es/map/maps.html#months=1#month=8#year=2022 (accessed on 23 March 2022)). The index

is calculated per month and per year. From this data, we calculate the annual average value of the index. These values were used to construct Figure 4.

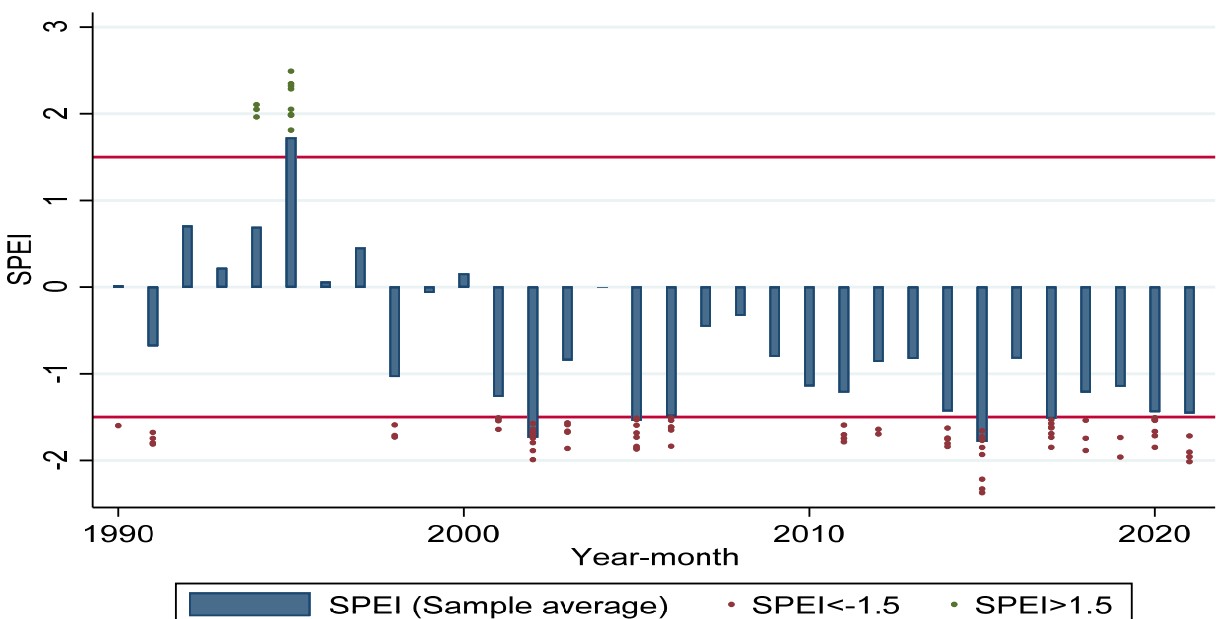

**Figure 4.** Monthly mean SPEI and SPEI shocks during the period 1990–2021 for Burkina Faso. Source: Own constructions based on data provided by the Global Drought Monitor: https://spei.csic.es/map/maps.html#months=1#month=1#year=2022 (accessed on 23 March 2022).

Figure 4 shows the evolution of the standard precipitation and evapotranspiration index for Burkina Faso. This index is widely used to quantify droughts. In addition, the index indicates the probability of losing water per cultivated area, and the zero value of the index marks the median, the negative value indicates drought conditions and the positive value shows wet conditions. Figure 4 indicates that the country has been characterized by declining soil moisture for a long time. The red horizontal lines in Figure 4 show the threshold levels, and when the SPEI is above the upper bound 1.5 the population is facing excess precipitation, while when the SPEI is below the lower bound −1.5 the population is facing drought shocks. The histograms show the evolution of the average SPEI over the entire country during the period 1990–2021. The green (red) scatterplots indicate the maximum (minimum) values of the SPEI recorded during the period 1990–2021 in specific areas. These scatterplots are local hot spots with SPEI values well above the mean values (the histograms), indicating that Burkina Faso has experienced severe and repeated drought shocks in certain areas of the country. Indeed, if the SPEI values lower than −1 show a light drought for the corresponding year, on the other hand SPEI values between −1 and −1.5 show a moderate drought situation and SPEI values higher than −1.5 indicate a severe drought situation.

Our drought scenarios are defined based on the changes in crop yield. Figure 1 shows that after the year 2000, Burkina Faso has experienced annual drought episodes in several localities of the country. The most severe droughts can be observed in the 2011–2012, 2013–2014 and 2018–2019 agricultural seasons. We measure the drought-related productivity shock through the proportional change in crop yields relative to the previous year. Figure 1 shows the interannual variations in crop yields between 2009 and 2019. With respect to the annual impacts of drought in Burkina Faso, The FAO data indicate that over the last ten years, the droughts of 2011–2012 resulted in reductions in crop yields of 13% for the 2011–2012 crop year, 15.8% for the 2013–2014 crop year and 2% for the 2018–2019 crop year. To take into account the uncertainty on the impact of drought on agricultural productivity, we use a stochastic shock model. This is because it is complex to model the

effects of droughts, as they can for years be intense, moderate and mild. According to Figure 1, each year Burkina Faso observes at least one drought episode, and due to the lack of estimated information on the probability of occurrence and the number of episodes per year, we consider a stochastic shock with a uniform probability distribution with 100 iterations. Finally, we capture the maximum, average and minimum mean impacts.

To implement our drought adaptation scenarios, we consider the occurrence of an intense drought. For example, ref. [24] analyzed the benefits and costs of four climate change adaptation measures in Burkina Faso: integrated soil fertility management, increased irrigation capacity, the adoption of improved seeds and a climate information system. They found that the most promising adaptation strategy was integrated soil fertility management, followed by increasing the irrigation capacity and the adoption of improved seeds, then finally the popularization of the climate information system. According to a World Bank study on the climate-smart investment plan for Burkina Faso, investments in water resources and irrigation are the best investments that show the highest increases in agricultural yields, followed by organic farming and water and soil conservation techniques [25]. Thus, in this study, we focus on the adoption of improved drought-tolerant seeds, integrated soil fertility management and irrigation development.

According to the World Bank's report on the climate-smart investment plan for the period 2018 to 2050 in Burkina Faso, an investment of $55 million over the period in the adoption of drought-tolerant crop varieties would increase agricultural productivity by 39%; investment of the same amount in water resources and irrigation would achieve an agricultural productivity increase of 56% in 2050; finally, the adoption of integrated soil management would allow for a productivity gain of 29% in 2050 with an investment of $55 million [25]. In this study, these estimates are used to implement our severe drought adaptation scenario. Finally, the results analysis is based on both short-term economic effects occurring in 2022 and long-term effects occurring in 2040. Our results analysis focuses on the effects of drought shocks on economic dynamics and the macroeconomic impacts on the outputs, employment and poverty according to household-headed gender.

## 4. Results and Discussion

### 4.1. Impact of Drought on Burkina Faso's Economy

4.1.1. Macroeconomic and Sectorial Impacts of Drought

Table 1 presents the macroeconomic and household results of stochastic simulations of three drought events on crop sectors. We define through the uniform probability of stochastic shock three impacts, whereby the maximum shock (most negative impact) is characterized as an intense drought, the medium impact is defined as a moderate drought and the less negative or positive impact is defined as a mild drought effect. Given the stochastic process used to generate the drought conditions over the period, each result is an average of the 100 iterations that define each of the three scenarios. The intensity of drought differs from year to year in each iteration of each scenario. For this reason, we present the results for short term (2022) and long term (2040). Therefore, we will present the average annual deviation from the baseline scenario as a percentage for the period 2018–2040.

As expected, drought leads to a reduction in GDP in all scenarios and the effects are more significant in the long run. The results show that the drought intensity results in a deterioration of macroeconomic and household welfare indicators. This result is consistent with those in CGE studies that focus on drought events [15,16,44,50]. These studies in other contexts also find reductions in GDP but of a higher magnitude than what we obtain in the case of Burkina Faso. Indeed, we find that an intense (severe) drought results in an average annual reduction in GDP of 3.0% in the short term and 3.3% in the long run relative to the baseline. However, moderate and mild drought events lead to reductions in GDP of 2.1% in the short run, 2.4% in the long run, 1.3% in the short term and 1.4% in the long term, respectively. The reduction in productivity in the agricultural sector makes the sector less attractive for investment, which reduces the capital stock in the agricultural sectors and the level of production. Given the decline in their level of production, each agricultural

sector reduces its intermediate consumption, and this negatively affects the other sectors. In addition, the decline in total production leads to reductions in total labor demand of 1.5% in the short term and 1.9% in the long term for an intense drought, 1.1% in the short term and 1.2% in 2040 for a moderate drought and 0.6% for a mild drought. Second, the decrease in production leads to a reduction in production supply and results in an increase in prices in the domestic market. Then, the intense, moderate and mild drought episodes lead to average increases of the consumer price index of 2.7% in 2022 and 3.5% in 2040 for an intense drought, 1.5% in the short term and 2.1% in the long term for a moderate drought and 0.5% in 2022 and 0.6% in 2040 for a mild drought.

**Table 1.** Impact on macroeconomic and household welfare variables for different categories of drought (annual change from BAU in %).

| | Short Term | | | Long Term | | |
|---|---|---|---|---|---|---|
| | **Intense** | **Moderate** | **Mild** | **Intense** | **Moderate** | **Mild** |
| Economic indicators | | | | | | |
| Real GDP | −3.0 | −2.1 | −1.3 | −3.3 | −2.4 | −1.4 |
| Total Investment | −4.6 | −1.8 | 0.7 | −4.2 | −1.1 | 1.5 |
| Consumer price index | 2.7 | 1.5 | 0.5 | 3.5 | 2.1 | 0.6 |
| Employment | −1.5 | −1.1 | −0.6 | −1.9 | −1.2 | −0.6 |
| Households' welfare indicators | | | | | | |
| Real household consumption | | | | | | |
| Rural households | −1.5 | −1.0 | −0.6 | −1.9 | −1.4 | −0.9 |
| Urban households | −3.4 | −2.4 | −1.5 | −3.8 | −2.7 | −1.4 |
| Food access per capita | | | | | | |
| Rural households | −4.1 | −3.0 | −2.1 | −4.5 | −3.2 | −1.7 |
| Urban households | −4.1 | −2.6 | −1.1 | −3.7 | −2.4 | −0.5 |
| Food availability per capita | | | | | | |
| Rural households | −10.4 | −7.3 | −4.6 | −11.1 | −7.5 | −4.0 |
| Urban households | −8.8 | −5.9 | −3.3 | −9.2 | −5.8 | −2.6 |

Source: Calculations based on the CGE model.

The CGE models also allow for a distinction between the impacts on rural and urban households. The drought leads to a decline in real consumption for all households. The results of the drought scenarios contained in Table 1 show that it is the urban households (i.e., the non-farming households) that are more affected in terms of their reduced real consumption. This result is consistent with the results found by [16] in the case of Malawi. Moreover, the result is explained by the fact that the net consumers of agricultural products are the households most vulnerable to rising food prices. However, rural (farm) households producing their own agricultural products and spending a large portion on self-consumption do not experience the price increase. The study by [80] showed that during the 1984–1985 drought, rural households in Burkina Faso resorted to self-insurance in the form of grain stock adjustments to smooth their consumption. In addition, the decline in non-farm wages and the increase in unemployment caused by the migration of farm workers to the urban economy, i.e., non-farm workers, due to the decline in economic opportunities in rural areas, contributed to the loss of income of existing non-farm workers and to the real consumption of urban households.

The effect on food security is particularly notable. Due to the decline in agricultural production, the per capita food availability decreases by 10.4%, 7.3% and 4.6% in the short term and by 11.1%, 7.5% and 4.0% in the long term, depending on the intensity of the drought for rural households. Similarly, the per capita supply of foodstuffs decreases by 8.8%, 5.9% and 3.3% in the short term and by 9.2%, 5.8% and 2.6% in the long term for urban households according to the nature of the drought. Similarly, as food prices increase and household incomes decrease, the access to food per capita decreases, and the situation

is relatively more pronounced in rural areas compared to urban areas, depending on the period and intensity of the drought.

### 4.1.2. Sectoral Results

The sectors of activity are affected differently according to their sensitivity to drought. Table 2 reports the impacts of different drought events on the production and supply. As expected, for the period 2018–2040, a reduction in crop yields due to droughts would cause a reduction in agricultural sector production of 9.5% in the short run and 10.2% in the long run for severe droughts, 6.7% in the short term and 6.9% in the long term for moderate droughts and 4.2% in the short term and 3.7% in the long term for mild droughts. Although we obtained these results in the specific context of Burkina Faso, they also align with the results from other CGE studies that focused on drought events (e.g., [15] for Mali and [16] for Malawi). The reduction in production in the agricultural sectors leads to reductions in the export demands for agricultural products of 28.0% in the short term and 29.1% in the long term for intense drought, 19.0% in the short term and 18.5% in the long term for the moderate drought impact and 10.6% in the short run and 8.2% in the long run for a light drought, with an increase in the imports of agricultural products. The activities most sensitive to drought shocks, i.e., cotton, sesame, fruits and vegetables, tubers and legumes, recorded a significant drop in production. Moderately sensitive crops such as maize, rice, millet and sorghum experienced a moderate decline in production. Activities that are less sensitive to the reduction in crop productivity, such as fonio production, livestock farming, forestry and fishing and hunting, showed a relatively modest decline in production.

**Table 2.** Impact on sectoral production for drought events (annual change from BAU in %).

| Sectors | Short Term | | | Long Term | | |
|---|---|---|---|---|---|---|
| | Intense | Moderate | Mild | Intense | Moderate | Mild |
| | | | Production | | | |
| Agriculture | −9.5 | −6.7 | −4.2 | −10.2 | −6.9 | −3.7 |
| Industry | −0.8 | −0.4 | 0.0 | −1.2 | −0.7 | −0.3 |
| Service | −1.1 | −0.9 | −0.6 | −1.4 | −1.1 | −0.8 |
| | | | Exports | | | |
| Agriculture | −28.0 | −19.0 | −10.6 | −29.1 | −18.5 | −8.2 |
| Industry | 0.3 | 0.8 | 1.3 | 0.8 | 1.3 | 1.8 |
| Service | −0.3 | 0.4 | 1.3 | −1.2 | −0.4 | 0.4 |
| | | | Imports | | | |
| Agriculture | 28.2 | 17.5 | 8.3 | 27.8 | 16.5 | 4.3 |
| Industry | −2.1 | −1.4 | −0.8 | −1.8 | −1.2 | −0.6 |
| Service | −4.8 | −3.0 | −1.5 | −4.3 | −2.2 | −0.2 |

Source: Calculations based on the CGE model.

The shock of droughts in the agricultural crop sector is transmitted to the non-agricultural sectors through factor reallocation, price changes and intermediate demands. The higher prices for agricultural products reduce the purchasing power of the consumers, and consequently negatively affect the demand for all non-agricultural products. Indeed, non-agricultural sectors that are strongly linked to agricultural sectors experience a decline in production. These branches include, for example, the food industry, the manufacture of beverages and tobacco, the textile and clothing industry, the production of chemical products and accommodation and catering activities. Industries that are not directly connected to the agricultural sectors, such as mining and soap and medicine manufacturing, experience an increase in production. The decline in the supply of agricultural products leads to an increase in the price of agricultural products used for intermediate consumption for the various activities, and consequently increases the cost of production. Thus, the outputs of the industry and the service sector decrease. However, the increase in production

in the mining, manufacturing and soap and medicine sectors leads to an increase in export demands and a reduction in imports of industrial products. The reduction in production in the different agricultural sectors has a direct impact on the exports of agricultural products, which are decreasing. The reduction in the demand for factors of production has a negative impact on the remuneration of workers in the various sectors.

### 4.1.3. Impacts of Drought on Income and Poverty

All economic agents are negatively affected by drought episodes, especially when the drought is intense. The income of the firms, which comes mainly from capital income, is affected by the decline in the use of capital in the industries. This decline in firm income leads to reductions in savings and direct tax payments. Households are facing a decline in income due to the reduction in income from capital, labor and transfers. The decline in household income is mainly driven by the decline in the income of poor and non-poor urban households. Faced with the decline in household income, households reduce their level of consumption, with urban households being the most affected. The government revenue declines due to the reduction in direct taxes on households and firms and also indirect taxes.

To assess the impacts of droughts on the evolution of poverty in Burkina Faso, we use the traditional FGT indicators used by [76]. We calculate these indicators for urban and rural areas and for male- and female-headed households. Table 3 summarizes the impacts of drought on poverty by residence and gender. In the baseline situation, we can observe that the poverty rates are higher among rural populations and among male-headed households. The drought causes increases in the poverty rate of 6.5% for an intense drought and 2.9% for a moderate drought and a decrease of 0.2% for a mild drought. Depending on the area of residence, the poverty rate is growing faster in urban areas than in rural areas.

**Table 3.** Impacts on poverty rates (annual change from BAU value in %).

| Population Groups | BAU | Short Term | | | Long Term | | |
|---|---|---|---|---|---|---|---|
| | | Intense | Moderate | Mild | Intense | Moderate | Mild |
| **Headcount Poverty (P0)** | | | | | | | |
| Areas | | | | | | | |
| -Urban | 13.2 | 18.4 | 14.8 | 12.2 | 62.1 | 31.1 | 10.9 |
| -Rural | 51.1 | 61.7 | 55.8 | 49.5 | 94.0 | 76.3 | 45.4 |
| **Household head gender** | | | | | | | |
| -Male | 42.3 | 51.8 | 46.3 | 40.8 | 86.5 | 65.9 | 37.2 |
| -Female | 32.8 | 38.8 | 35.3 | 31.6 | 80.0 | 52.9 | 30.0 |
| Population | 41.4 | 50.6 | 45.3 | 39.9 | 85.8 | 64.7 | 36.6 |
| **Poverty gap (P1)** | | | | | | | |
| Areas | | | | | | | |
| -Urban | 3.3 | 4.9 | 3.9 | 3.1 | 23.8 | 9.1 | 2.7 |
| -Rural | 15.3 | 20.5 | 17.3 | 14.4 | 52.1 | 31.0 | 12.9 |
| **Household head gender** | | | | | | | |
| -Male | 12.4 | 16.8 | 14.1 | 11.6 | 45.5 | 25.9 | 10.4 |
| -Female | 10.9 | 14.1 | 12.1 | 10.4 | 38.5 | 21.1 | 9.5 |
| Population | 12.2 | 16.5 | 13.9 | 11.5 | 44.8 | 25.4 | 10.3 |
| **Poverty severity (P2)** | | | | | | | |
| Areas | | | | | | | |
| -Urban | 1.3 | 1.9 | 1.5 | 1.1 | 11.8 | 3.8 | 1.0 |
| -Rural | 6.3 | 9.0 | 7.3 | 5.8 | 32.4 | 15.5 | 5.1 |
| **Household head gender** | | | | | | | |
| -Male | 5.0 | 7.3 | 5.8 | 4.6 | 27.5 | 12.7 | 4.0 |
| -Female | 4.8 | 6.6 | 5.5 | 4.5 | 22.9 | 10.8 | 4.0 |
| Population | 5.0 | 7.2 | 5.8 | 4.6 | 27.1 | 12.5 | 4.0 |

Source: Calculations based on the micro-simulation model.

Table 3 shows that while all household groups are experiencing an increase in the number of poor, rural households are the most affected. Urban households account for only about 30% of the total population and a smaller share of the poor population. In fact, rural households contribute to more than 90% of the national poverty. Thus, changes in poverty among these households largely dictate the national poverty level following the droughts. The results show a significant increase in poverty among rural households. Thus, the poverty rate at the national level would increase from 41.4% in the reference year to 50.6% in 2022 and 85.8% in 2040 in a severe drought situation. Under a moderate drought, the poverty rate would increase from 41.4% to 45.3% in the short term and to 64.7% in the long term. In the context of a mild drought, the poverty rate decreases at the national level because the situation does not produce significant inflation, thereby reducing the purchasing power of households. Depending on the area of residence, a significant proportion of rural households fall below the poverty line. For a severe drought, for example, the poverty rate for the rural households would rise from 51.1% in the reference year to 61.7% in the short term and 94.0% in the long term, compared with a poverty rate of 13.2% for the urban households at the baseline, which would rise to 18.4% in the short term and 62.1% in the long term.

Focusing on the gender dimension of poverty, Table 3 shows that agricultural droughts affect the welfare of both male and female-headed households. The microsimulation results show that male-headed households are more affected by drought than female-headed households. This result can be explained by the fact that according to the latest survey on household living conditions in Burkina Faso in 2018, 90.6% of households are headed by men compared to 9.4% of households being headed by women. In addition, 50.5% of male-headed households are female versus 49.5% being male, and 64% of female-headed households are female versus 36% being male. Indeed, compared to the baseline situation, the poverty rate of male-headed households increases and jumps to 42.3% in the reference year to 51.8% in 2022 and 86.5% in 2040 compared to female-headed households, for which the poverty rate increases from 32.8% in the initial situation to 38.8% in the short term and to 80.0% in the long run in the context of intense drought. Similarly, the microsimulation results show that in the moderate drought context, poverty rate remains high compared to the reference year. Due to the income increase in the mild drought context and the light increase in commodity prices, the poverty rate declines (Table 3). If we take into account the weight of the female population, the results seem to confirm that droughts worsen the standard of living for women more than men, and those living in rural areas may be the most affected. In addition, the results show an increase in the depth of poverty by the area of residence and by gender through the P1 indicator and also an increase in the severity of poverty through the P2 indicator.

*4.2. Drought Adaptation Strategies Impact Socioeconomic Indicators*

The final step in our analysis of the impact of agricultural drought on the agricultural sector is to identify the scope for adaptation measures in Burkina Faso. Several adaptation strategies are identified in Burkina Faso's climate change adaptation plan. In this study, we focus on three adaptation options capable of improving the productivity of the agriculture sector: an expansion of the adoption of drought-tolerant crops, the adoption of integrated soil management and the expansion of the irrigation capacity. As noted above, the [25] finds that an investment of $55 million in the adoption of drought-tolerant crop varieties yields an average 39% increase by 2050, while investment in water resource management and irrigation yield an average increase of 56% by 2050 and investment in integrated soil management yields a 29% increase by 2050.

Given that the objective of the selected adaptation measures is to improve the productivity of the agricultural sector, we introduce a gradual increase in agricultural productivity using the adaptation strategy and determine the threshold at which the negative impact is neutralized. Table 4 presents the impacts of the adoption of drought-tolerant crop varieties on growth, food security and the poverty rate by household head gender. Indeed, with a

wide use of drought tolerant crop varieties, the negative impacts on the GDP, investment, price, employment and urban household consumption could be reduced progressively. We find by 2035 a 21% increase in agricultural crop productivity, where the intense drought's negative impacts on GDP and employment become positive and increase by 0.2% and 0.1%, respectively.

**Table 4.** Impacts of drought-tolerant crop adoption on growth and food security.

|  | **2022** | **2025** | **2030** | **2035** | **2040** |
|---|---|---|---|---|---|
| Agricultural productivity change | 6% | 9% | 15% | 21% | 27% |
| Real GDP | −2.6 | −2.0 | −0.8 | 0.2 | 1.3 |
| Total Investment | −1.7 | −0.8 | 0.3 | 1.1 | 1.9 |
| Consumer price index | 2.1 | 1.7 | 0.9 | 0.1 | −0.7 |
| Employment | −1.4 | −1.1 | −0.5 | 0.1 | 0.6 |
| Food access per capita |  |  |  |  |  |
| Rural households | −3.8 | −2.8 | −1.1 | 0.6 | 2.3 |
| Urban households | −3.0 | −2.1 | −0.6 | 0.9 | 2.4 |
| Food availability per capita |  |  |  |  |  |
| Rural households | −9.0 | −6.7 | −2.6 | 1.6 | 5.8 |
| Urban households | −7.1 | −5.2 | −1.9 | 1.5 | 5.0 |
| Poverty rate by household head gender |  |  |  |  |  |
| Male-headed households | 37.8 | 32.0 | 22.0 | 13.9 | 8.6 |
| Female-headed households | 30.2 | 26.0 | 20.1 | 13.5 | 9.6 |

Source: Calculations based on the CGE model and micro model.

The results indicate that the adoption of new crop varieties is a profitable investment strategy in the long run that will increase the real GDP by 1.3% by 2040. Replacing old varieties with modern ones increases the agricultural production and reduces prices in the long run. This increases the purchasing power of households. Urban households benefit more than rural households. As a result, due to the increase in agricultural production, the per capita food availability increases by 5.8% for rural households and 5.0% for urban households in the long term. Similarly, the decreases in agricultural commodity prices and household income increase the purchasing power of households, leading to an increase in the food access indicator for both urban and rural households from 2035, with about 21% agricultural productivity growth. This strategy is beneficial because the gains are significant; however, the development of drought-tolerant crops must take into account the agroecological zones in Burkina Faso. In addition, the success of this policy is linked to the implementation of other programs or policies such as sustainable agriculture practices that integrate technical assistance (training, access to fertilizers) and agrometeorological assistance. Finally, compared to the poverty rate in the case of a severe drought, the improved income and lower prices for goods and services on the domestic market lead to a reduction in the incidence of poverty. The reduction in poverty is faster in male-headed households than in female-headed households (Table 4).

In addition, the adoption of the integrated soil management technique contributes to the improvement of economic and social indicators and to the neutralization of the impacts of intense drought. However, the gains in terms of employment generation, growth and food security are slow. Given the low rate of productivity increase with this technology, it is not until 2040 that an increase in agricultural productivity of 20% is seen, reversing the negative effect of drought. Thus, an increase of 20% in 2040 in the productivity of the agricultural sector following the adoption of integrated soil management offsets the negative effects of drought on the added agricultural value and on real household consumption and leads to an increase in real GDP of 0.1%. The adoption of integrated land management improves the food security situation through increased agricultural production and also through lower prices and improved household income. Thus, the per capita access to food and per capita food availability increase for rural households and for urban households. This policy plays an important role in soil fertilization and restoration,

and is an endogenous strategy that could be easily adopted by farmers. This strategy is also effective and easy to popularize but generates fewer benefits than the adoption of drought-tolerant crops. Its impact on poverty reduction in both male- and female-headed households is slow (Table 5).

**Table 5.** Impacts of integrated soil management adoption on growth and food security.

|  | 2022 | 2025 | 2030 | 2035 | 2040 |
|---|---|---|---|---|---|
| Productivity growth change | 4% | 7% | 11% | 16% | 20% |
| Real GDP | −3.0 | −2.5 | −1.6 | −0.8 | 0.1 |
| Total Investment | −2.1 | −1.2 | −0.4 | 0.2 | 0.8 |
| Consumer price index | 2.4 | 2.2 | 1.5 | 0.8 | 0.2 |
| Employment | −1.6 | −1.3 | −0.9 | −0.4 | 0.0 |
| Food access per capita |  |  |  |  |  |
| Rural households | −4.3 | −3.6 | −2.3 | −0.9 | 0.3 |
| Urban households | −3.4 | −2.6 | −1.5 | −0.5 | 0.6 |
| Food availability per capita |  |  |  |  |  |
| Rural households | −10.0 | −8.3 | −5.3 | −2.2 | 0.9 |
| Urban households | −7.9 | −6.5 | −4.1 | −1.7 | 0.9 |
| Poverty rate by household head gender |  |  |  |  |  |
| Male-headed households | 38.0 | 32.2 | 22.5 | 14.2 | 9.1 |
| Female-headed households | 30.3 | 26.0 | 20.5 | 13.6 | 10.2 |

Source: Calculations based on the CGE model and micro model.

Finally, the results indicate that the increased irrigation capacity could fully offset the negative effects of the intense drought through its capacity to increase agricultural productivity (Table 6). Indeed, an increase in crop productivity of 18.77% through water and irrigation management leads to the neutralization of the effects of agricultural drought. This technology allows a faster reduction in drought effects than the two previous technologies. In this scenario, achieving a 39% increase in agricultural productivity through increased irrigation capacity would be sufficient to revive the agricultural sector from the drought. Among the adaptation options, this option is more profitable, with an increase in GDP of 3.1% in 2040. The expansion of the irrigation capacity helps to improve economic and social indicators. The real consumption of urban households increases by 3.5% by 2040, while that of rural households increases by 1.3% in the long term. In addition, the development of the irrigation capacity is a tool for improving food security. These results are comparable to those found by [50] in the case of Niger and by [15] in the case of Mali in the long run. The application of irrigation is very important for Burkina Faso, not only during the dry season, but also to supplement low rainfall during the rainy periods. Thus, the use of irrigation is a relevant alternative approach because the country has developed only 33% of its irrigation capacity on an irrigable area estimated at 233,500 hectares [81]. This option for adapting to climate change is to be encouraged because it primarily relates to cash crops, which account for a large share of the agricultural value added, and it is relatively less costly compared to the productivity gains recorded. Finally, it is also a strategy that allows for more poverty reductions in male-headed households than in female-headed households (Table 6).

**Table 6.** Impacts of irrigation investment on growth, food security and poverty.

|  | 2022 | 2025 | 2030 | 2035 | 2040 |
|---|---|---|---|---|---|
| Agricultural productivity change | 8% | 14% | 22% | 31% | 39% |
| Real GDP | −2.0 | −1.1 | 0.4 | 1.8 | 3.1 |
| Total Investment | −1.1 | 0.1 | 1.4 | 2.6 | 3.8 |
| Consumer price index | 1.7 | 1.1 | 0.1 | −0.9 | −1.7 |
| Employment | −1.1 | −0.6 | 0.1 | 0.8 | 1.3 |
| Food access per capita |  |  |  |  |  |
| Rural households | −3.1 | −1.6 | 0.8 | 3.2 | 5.5 |
| Urban households | −2.3 | −1.0 | 1.2 | 3.3 | 5.4 |
| Food availability per capita |  |  |  |  |  |
| Rural households | −7.2 | −3.8 | 2.1 | 8.2 | 14.2 |
| Urban households | −5.7 | −2.9 | 2.0 | 7.0 | 12.1 |
| Poverty rate by household head gender |  |  |  |  |  |
| Male-headed households | 37.4 | 31.7 | 21.5 | 13.7 | 8.5 |
| Female-headed households | 30.2 | 25.4 | 19.9 | 13.5 | 9.1 |

Source: Calculations based on the CGE model and micro model.

## 5. Conclusions

Burkina Faso's climatic condition is marked by high exposure to drought. This study uses a dynamic CGE model combined with a microsimulation model to assess the impacts of drought episodes and adaptation strategy options on the economy and their consequences on poverty between 2018 and 2040. By simulating three drought scenarios over a 22-year period, we first show how the impacts on agricultural production of each mild, moderate and severe drought the country experiences contribute to deteriorating economic performance and significantly damaged household welfare. In addition, our paper describes how the negative impacts will be exacerbated by the increase in the numbers of intense drought events with present and future climate changes. However, we find that there will be a number of options for Burkina Faso to cope with the adverse and changing climatic conditions if the country is to proceed with the implementation of adaptation options, such as the adoption of drought-tolerant crop varieties, the adoption of integrated soil management and the development of water resource management and irrigation.

The results show that the different degrees of drought have negative effects on agricultural and non-agricultural production. Thus, the different drought episodes degrade the economic performance of the country as the GDP decreases, employment demands are reduced and real household consumption falls. The results show that food availability and access to food are reduced. Additionally, the drought episodes increase the poverty rate.

Specifically, the results show that rural farmers are the most vulnerable to the decline in farm income and the increase in poverty. Urban households also experience increased poverty due to rising food prices and falling off-farm wages. Indeed, the disruption of the supply chain due to drought indirectly leads to reduced production in downstream agribusinesses and upstream services. Furthermore, the results show that male-headed households are the most affected by agricultural droughts, because the number of poor people is increasing rapidly in this group of households. Given the size of the female population, women are bearing the brunt of the costs of drought. Finally, we show that Burkina Faso has some options to cope with the current agricultural droughts through the adoption of new drought-tolerant crop varieties, the popularization of integrated soil management and the development of irrigation capacities.

The results provide information on Burkina Faso's vulnerability to agricultural drought and the country's capacity to adapt. However, our results do not take into account regional differences in drought sensitivity. Our analysis is limited to economic and poverty impacts without considering conflict, health risks and migration.

Finally, the adaptation strategies considered in this study can neutralize the severe drought events. The results show that the development of irrigation seems to have better results in terms of poverty reduction, followed by the adoption of drought-tolerant cereal

varieties and finally the adoption of integrated soil management. However, this study has important limitations. First, our CGE model does not fully address the issue of uncertainty about the nature of droughts and the cost–benefit analysis. In addition, we did not take into account the range of adaptation strategies available in Burkina Faso. Additionally, we did not consider the issue of the financing mechanism, although this is an important issue in the context of the constrained fiscal space in Burkina Faso. Finally, as a low-income country, Burkina Faso's agriculture production is sensitive to climate change and the farmers have low adaptive capacities. In addition, Burkina Faso is currently facing multiple crises related to security, institutional issues and the COVID-19 pandemic, which complicate the implementation of agricultural development programs and policies. In this context, the policy question is how to find a development plan that can resolve the cyclical crises and promote the development of agricultural production for sustainable pro-poor economic growth.

**Funding:** This research was funded by African Economic Research Consortium (AERC) grant number RC21587 and The APC was funded by African Economic Research Consortium (AERC).

**Institutional Review Board Statement:** Not applicable.

**Informed Consent Statement:** Not applicable.

**Data Availability Statement:** The dataset used and analyzed during the current study is available from the corresponding authors on reasonable request.

**Acknowledgments:** This study has benefited from technical and financial assistance from the African Economic Research Consortium (AERC), The authors are grateful for the constructive comments and advice given by all participants in the three AERC workshops. I would like to thank Gansonre Soumaïla for their excellent research assistance. The views expressed in this article are those of the author and do not necessarily express the views of the African Economic Research Consortium.

**Conflicts of Interest:** The authors declare no conflict of interest.

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
