# Peer review of "Drought Impacts on the Crop Sector and Adaptation Options in Burkina Faso: A Gender-Focused Computable General Equilibrium Analysis"

_sustainability, doi:10.3390/su142315637_

Round 1

Reviewer 1 Report

Abstract: It is not clear in current form as not giving any message. It should have 

1. Rationale

2. Method

3. Results in numbers

4. Recommendations for all

Model Calibration and evaluation section is missing.

Author Response

Point 1: Abstract: It is not clear in current form as not giving any message. It should have 

  1. Rationale ; 2. Method ; 3. Results in numbers ; 4. Recommendations for all ; Model Calibration and evaluation section is missing.

.

Response 1: We have responded to your comments on reorganising the summary. You will find that the summary has undergone major changes to provide more clarity on the context of the study, the method used, the results and the recommendations. The model calibration and evaluation is in section 3 of the report.

Reviewer 2 Report

In attachment

Author Response

Point 1: Drought impacts on crops sector and adaptation options in 2 Burkina Faso: a gender focused computable general equilibrium analysis

Dear Author,

the work is very interesting and current. It is especially important that gender equality is also represented. However, he suggests a few more minor corrections. In the Introduction, it would be good to refer to the reference IPCC 2014 (WGII); for Adaptation). The IPCC Phyisical sciences basis reference should be cited correctly (it is misquoted in the references). It would be good to provide a map of the geographical position of Burkina Faso in relation to Africa, for future readers who do not know where it is. In conclusion, it would be necessary to state whether there are sufficient amounts of water in Burkina Faso to enable the irrigation of agricultural crops. After these corrections, I can recommend the paper for publication.

Best regards.

Response : We thank the reviewer for these pertinent comments which will help to improve the quality of the paper.

As suggested by the reviewer, we did quote the IPCC 2014 reference (WGII) and also, The IPCC (2013) Phyisical sciences basis reference is cited correctly (it is quoted in the references). We have added a section 2 for the presentation of the scope of the study with a supporting map. The other suggestions are integrated in the report.

Reviewer 3 Report

The manuscript presents a very useful case study and the results are in good agreement with the mentioned methodology. The Introduction section must be reviewed. Apart from compiling key studies, a critical approach to the literature is required. It is true that the authors address key concepts, but this is not good enough for a solid, serious research paper. Some important concepts are discussed but the reader doesn’t know why they are relevant or the connection between them.

Author Response

Point 1: The manuscript presents a very useful case study and the results are in good agreement with the mentioned methodology. The Introduction section must be reviewed. Apart from compiling key studies, a critical approach to the literature is required. It is true that the authors address key concepts, but this is not good enough for a solid, serious research paper. Some important concepts are discussed but the reader doesn’t know why they are relevant or the connection between them.

Response: As proposed by the rapporteur, the introduction section has been revised. Suggestions in the literature have been taken into account in the revision of the introduction. All substantive comments have been addressed in the text.

Reviewer 4 Report

In general terms, the article is of importance for Burkina Faso from the point of view of development and adaptation to climate change. The work covers climate issues with socio-economic components that aggravate the impact of droughts on society. They also attempt to provide specific solutions to the problems associated with droughts.

However, the manuscript needs to be revised for publication. The main problems I find are the difficult readability of the paper and the wrong timing in the presentation of the information, both before (in the introduction) and during the paper itself (methodology and results).

In addition, there is a lack of graphs and tables to facilitate the understanding of the work so that its replicability is guaranteed. Below, I highlight some factors to be taken into account when improving the manuscript.

Abstract:

I recommend separating the ideas from lines 8-11 in the abstract to make it more concise and easier to read.

Revise the English: verb tenses can be improved (line 13-14).

Explain the methodology in a more specific and not so general way. Also highlight some numerical results and some more specific ones in the abstract.

Introduction:

- In the first paragraph, I recommend a better threading of the ideas. It talks in a general way about climate change, then it specifies a problem about droughts, and then goes back to other issues such as floods and storms. In addition, the role of gender in the drought problem is suddenly touched upon without having been introduced beforehand. Specific references are missing for some statements, while there are several in other sentences that could not be used or specify specific information from those quotes.

- In the second paragraph, better link drought events with impacts on crops and nutrition.

- Line 52 talks about gender-focused objectives. Before this I recommend introducing the topic and the drought-impact/gender issue. Since references to this sub-theme are only touched upon in line 71.

- Can reference number 22 be updated or reinforced?

- Are there any extra effects now or in the coming years due to the impact of COVID-19, armed conflicts, free trade agreements, etc.? Not to focus the study, but to complement that, even if it is the climate that triggers a problem, the multifaceted conditions in Burkina Faso create very serious situations socio-economically speaking. And that in which this study focuses on gender and climate.

- Paragraph 103-115 is very general after having discussed a lot of data specific to the study area.

- To introduce the gap in a better way and why the chosen model is the best choice for the analysis first.

- CGE in line 122 is written for the first time without introducing it previously.

- It could be improved to explain the specific objectives and how they will be addressed. Who is the study aimed at?

Material and Methods:

- I recommend introducing a map with the location of the country under study, together with some data of interest: rainfall, temperature, rain-fed farming areas, major settlements, or any other information that the authors consider important.

- The explanation of the model could be supported with flow charts, graphs and previous studies that justify the assumptions. I also recommend introducing clear limitations to the study.

- References to the selected drought indices are missing. What data were used to calculate them, and why are these indices used and not others?

- Figure 3 is named without going through Figure 2. Furthermore, only one figure has been uploaded. Figure 1 can be improved, it is not understandable by itself.

- I recommend integrating tables and graphs for the information provided to make the document easier to read. Especially about the variables related to the model and how to interpret the results.

- The exact procedure and the assignment of values to the model is not clear. I recommend clarifying the methodology to make it easily replicable. Too much text makes it difficult to follow the process.

Results and Discussion:

- I recommend adding figures to facilitate and better represent the results obtained.

- In the text the tables are in disorder and not all of them are named.

- The results associated with section 3.2 are not clearly detailed in the methodology. The results themselves are inconclusive without further discussion or the use of other statistics to confirm the use of the model.

- Do the study scales have an influence?

- The data used correspond to sample or total data. This fact could affect the discussion of the problem.

Conclusions

- I recommend making the conclusions more specific.

- Relate the conclusions to the specific objectives of the introduction.

Author Response

Point 1: In general terms, the article is of importance for Burkina Faso from the point of view of development and adaptation to climate change. The work covers climate issues with socio-economic components that aggravate the impact of droughts on society. They also attempt to provide specific solutions to the problems associated with droughts.

However, the manuscript needs to be revised for publication. The main problems I find are the difficult readability of the paper and the wrong timing in the presentation of the information, both before (in the introduction) and during the paper itself (methodology and results).

In addition, there is a lack of graphs and tables to facilitate the understanding of the work so that its replicability is guaranteed. Below, I highlight some factors to be taken into account when improving the manuscript.

Response 1: We would like to thank the reviewer for these pertinent comments.

We have provided responses to the formal and substantive comments raised by the reviewer. The whole text has been revised to correct all the editorial comments made directly in the text. All substantive comments have been addressed in the text.

Point 2: Abstract:

I recommend separating the ideas from lines 8-11 in the abstract to make it more concise and easier to read.

Revise the English: verb tenses can be improved (line 13-14).

Explain the methodology in a more specific and not so general way. Also highlight some numerical results and some more specific ones in the abstract.

Response 2: As suggested by the reviewer, we have revised the abstract to give greater clarity. In the revision, we have taken into account the suggested comments.

Point 3: Introduction:

- In the first paragraph, I recommend a better threading of the ideas. It talks in a general way about climate change, then it specifies a problem about droughts, and then goes back to other issues such as floods and storms. In addition, the role of gender in the drought problem is suddenly touched upon without having been introduced beforehand. Specific references are missing for some statements, while there are several in other sentences that could not be used or specify specific information from those quotes.

- In the second paragraph, better link drought events with impacts on crops and nutrition.

- Line 52 talks about gender-focused objectives. Before this I recommend introducing the topic and the drought-impact/gender issue. Since references to this sub-theme are only touched upon in line 71.

- Can reference number 22 be updated or reinforced?

- Are there any extra effects now or in the coming years due to the impact of COVID-19, armed conflicts, free trade agreements, etc.? Not to focus the study, but to complement that, even if it is the climate that triggers a problem, the multifaceted conditions in Burkina Faso create very serious situations socio-economically speaking. And that in which this study focuses on gender and climate.

- Paragraph 103-115 is very general after having discussed a lot of data specific to the study area.

- To introduce the gap in a better way and why the chosen model is the best choice for the analysis first.

- CGE in line 122 is written for the first time without introducing it previously.

- It could be improved to explain the specific objectives and how they will be addressed. Who is the study aimed at?

Response 3: All the comments raised have been taken into account point by point in the revision of the introduction. Indeed, we have responded to the formal and substantive comments raised by the reviewer. The whole text of the introduction has been revised to correct all editorial comments made directly in the text and are in track changes. All substantive comments have been addressed in the text.

Point 4: Material and Methods:

- I recommend introducing a map with the location of the country under study, together with some data of interest: rainfall, temperature, rain-fed farming areas, major settlements, or any other information that the authors consider important.

- The explanation of the model could be supported with flow charts, graphs and previous studies that justify the assumptions. I also recommend introducing clear limitations to the study.

- References to the selected drought indices are missing. What data were used to calculate them, and why are these indices used and not others?

- Figure 3 is named without going through Figure 2. Furthermore, only one figure has been uploaded. Figure 1 can be improved, it is not understandable by itself.

- I recommend integrating tables and graphs for the information provided to make the document easier to read. Especially about the variables related to the model and how to interpret the results.

- The exact procedure and the assignment of values to the model is not clear. I recommend clarifying the methodology to make it easily replicable. Too much text makes it difficult to follow the process.

Response 4:  As suggested by the reviewer, we have introduced a section 2 to present Burkina Faso, the geographical position, climate and other information relevant to the study.

In section 3, we have presented the analysis framework in a graph and also a figure is used to illustrate the modelling process.

All substantive comments have been addressed in the text.

Point 5: Results and Discussion:

- I recommend adding figures to facilitate and better represent the results obtained.

- In the text the tables are in disorder and not all of them are named.

- The results associated with section 3.2 are not clearly detailed in the methodology. The results themselves are inconclusive without further discussion or the use of other statistics to confirm the use of the model.

- Do the study scales have an influence?

- The data used correspond to sample or total data. This fact could affect the discussion of the problem.

Response 5: In organising the text, we felt that tables could facilitate the presentation of the results rather than graphs which would be more complex. All tables in the text have been revised and correctly named to facilitate understanding. The data we use are national level data. In the methodology section we have integrated the specifications of all the indicators used in the results. The whole text has been revised to take into account the substantive and formal comments.

Point 6: Conclusions

- I recommend making the conclusions more specific.

- Relate the conclusions to the specific objectives of the introduction.

Response 6: The conclusion section has been revised to take into account the comments on substance and form. All changes are in track changes.

Reviewer 5 Report

The manuscript entitled „Drought impacts on crops sector and adaptation options in Burkina Faso: a gender focused computable general equilibrium analysis” presents interesting study on negative effect of drought on agriculture and other sectors.

However there are some drawbacks which should be corrected.

1) Most of the results in the study are based on the model for which following reference is cited:

B. Decaluwé, A. Lemelin, V. Robichaud, and H. Maisonnave, pep -1- t the PEP standard single-country, recursive dynamic CGE model, vol. 0, no. May 2012. Québec (Canada): Partnership for Economic Policy, 2013

The publication is easy available. Could you cite other reference or provide more details aboout the model? What input data were used for the calculation, yearly data or other?

2) More details about sources of the data should be provided. The statistical inpust data were based on sample or total population?

3) It is not clear if the results presented in all the Tables (1-5) are presented in percentages or percentage point. The result assume that effect of drought is for subsequent years and is linear?

4) It would be good if the results in this study were compared to other similar results, for example other country. The discussion is quite poor becuase it focus on Burkina Faso and there are no more comprehensive comparison to the similar results for different countries.

Author Response

Point 1: The manuscript entitled „Drought impacts on crops sector and adaptation options in Burkina Faso: a gender focused computable general equilibrium analysis” presents interesting study on negative effect of drought on agriculture and other sectors.

However there are some drawbacks which should be corrected.

1) Most of the results in the study are based on the model for which following reference is cited:

  1. Decaluwé, A. Lemelin, V. Robichaud, and H. Maisonnave, pep -1- t the PEP standard single-country, recursive dynamic CGE model, vol. 0, no. May 2012. Québec (Canada): Partnership for Economic Policy, 2013

The publication is easy available. Could you cite other reference or provide more details aboout the model? What input data were used for the calculation, yearly data or other?

Response 1: In the methodology section, paragraph 4, we have examples of recent studies already published that have used the same model.

Point 2:  More details about sources of the data should be provided. The statistical inpust data were based on sample or total population?

Response 2: Section 3.2 provides an overview of the data sources used in this study. The data used covers the whole country.

Point 3: It is not clear if the results presented in all the Tables (1-5) are presented in percentages or percentage point. The result assume that effect of drought is for subsequent years and is linear?

Response 2: The results presented in the text are in percentages compared to the baseline situation. Three types of drought were determined in this study, mild drought, moderate drought and severe drought. The results depend on the intensity of the drought.

Point 4: It would be good if the results in this study were compared to other similar results, for example other country. The discussion is quite poor becuase it focus on Burkina Faso and there are no more comprehensive comparison to the similar results for different countries.

Response 2: These comments are reflected in the text. We have tried to compare the results we have with the results of other studies especially in the African context.

Round 2

Reviewer 5 Report

The manuscript was changed according all my comments.

Some technical changes are necessary. Quality of Fig. 2 and 3 is not suffcient. It should be improved.

Author Response

Point 1: Some technical changes are necessary. Quality of Fig. 2 and 3 is not suffcient. It should be improved.

Response 1: Thanks to the reviewer for the remark, we have taken figure 2 and 3. These figures are placed in the text.
